# Trends in Transfemoral Aortic Valve Implantation Related Thrombocytopenia

**DOI:** 10.3390/jcm11030726

**Published:** 2022-01-29

**Authors:** Haitham Abu Khadija, Gera Gandelman, Omar Ayyad, Lion Poles, Michael Jonas, Offir Paz, Jacob George, Alex Blatt

**Affiliations:** 1Heart Center, Kaplan Medical Center, Rehovot 76100, Israel; haithamab1@clalit.org.il (H.A.K.); Gera_G@clalit.org.il (G.G.); omeray@clalit.org.il (O.A.); Lion_P@clalit.org.il (L.P.); MichaelYo2@clalit.org.il (M.J.); ofirrpa@clalit.org.il (O.P.); Kobige@clalit.org.il (J.G.); 2Mediciene Faculty, Hebrew University, Jerusalem 91120, Israel

**Keywords:** aortic valve, thrombocytopenia, transfemoral aortic valve replacement, TAVI, trends, contemporaneous

## Abstract

Background: TAVI related thrombocytopenia (TAVI-rTP) is still very common. The aim of this study was to compare the incidence, characteristics and impact of reduced platelet counts (RPC) after TAVI between an earlier and contemporary period. Methods: the patients enrolled were those experiencing severe symptomatic aortic stenosis who underwent TAVI between January 2010 and December 2019. The exclusion criteria were no available blood tests and periprocedural death. Results: 334 patients (mean age 81.9 ± 6.7 years) were enrolled. For the earlier period, the mean RPC was 33 ± 15%, and in the contemporary period (2016–2019) it was 26 ± 14%. In the early group, we found that 62% of the patients had decreased platelet counts of more or equal to 30% in comparison to 33% in the contemporary period. The time of the procedure and the amount of the contrast that had been used in the later period were associated with significant RPCs (*p* value = 0.002 and 0.028, respectively). An RPC of 30% or more was associated with the increased risks of life-threatening bleeding, vascular complications and death within 30 days. Conclusion: contemporary TAVI-rTP continued to be a common phenomenon in our cohort. However, severe thrombocytopenia was significantly less frequent. An RPC of 30% or more is associated with a poor 30-day outcome.

## 1. Introduction

Transcatheter aortic valve implantation (TAVI) related thrombocytopenia (TAVI-rTP) is a common periprocedural finding [1,2,3]. Flaherty et al. coined the concept that TAVI-rTP may be a “universal and virtually inevitable” phenomenon [4]. Due to better approaches and improved technologies, the indications for TAVI over the past decade have been extended and are now not only for patients with prohibitive or high-operative risk [5,6,7]. Additionally, the TAVI procedure has evolved to be simpler and safer [8,9]. Despite these improvements, studies show that a decrease in platelet count appears in nearly 90% of the treated patients [4,10,11], with different degrees of severity, and has been associated with worse clinical outcomes [11,12,13,14].

The current explanation for the TAVI-rTP mechanism is platelet activation and a systemic inflammatory response [2,15,16,17]. Even though TAVI-rTP has been thoroughly described, a comprehensive understanding of this phenomenon needs further investigation. To the best of our knowledge, there are no studies investigating the impact of contemporary TAVI approaches on this phenomenon compared with earlier experiences. The objectives of the present study were to compare the incidence, characteristics and impact of reduced platelet count (RPC) after TAVI between an earlier and contemporary period.

## 2. Methods

### 2.1. Patient Population

Consecutive patients were retrospectively included if they had severe symptomatic aortic stenosis and had underwent transfemoral TAVI at our center between January 2010 and December 2019. Patients were excluded if they had suffered periprocedural death (up to 72 h after TAVI) and also if their post-TAVI platelet counts were not available. The medical team at the time of operation had full discretion when it came to the type and size of the valves. The choice was between Sapien balloon-expandable (BEV), Sapien XT, S3 valves (Edwards Lifesciences, Irvine, CA, USA) or self-expanding (SEV) Corevalve, Evolut R or Evolute PRO (Medtronic, Inc., Minneapolis, MN, USA) valves. Those treated with other valves were very few in number and were also excluded from the analysis. A percutaneous approach was used for transfemoral vascular access and closure and a safety wire technique along with a Prostar XL (Abbott Vascular, Redwood City, CA, USA) vascular closure device or Manta were used. The calculation of the procedure duration was from “skin to skin”, with time 0 being the arterial blood pressure opening from the accessory support access and the concluding time was considered to be the accessory support access closure. The first line approach was by using local anesthesia with conscious sedation. Unfractionated heparin was given to all patients for maintaining minimum active clotting times of over 250 s after femoral sheet insertion. During vascular closure, protamine (1 mg for each 100 U of heparin, maximal dose 50 mg) was given on a needs basis. The patients were recommended aspirin before TAVI. Dual-antiplatelet treatment with clopidogrel 75 mg and aspirin 100 mg was initiated a day before the procedure and thereafter for half a year, except for those patients requiring chronic oral anticoagulation.

The clinical outcomes, procedural data and baseline characteristics were collected. Laboratory analyses were performed at the following time points: before the TAVI procedure, daily during the stay in postprocedural ICU and at the discretion of the cardiology ward physician. All data were retrospectively collected. The standard follow-up included visits thirty-days and half a year after discharge from the hospital and were performed on site.

We divided our study population into two groups, the “early TAVI era group”, which included patients that were implanted with first generation valves, i.e., Sapien, Sapien XT and Corevalve. The second group, the “contemporaneous TAVI era” group, employed the newer generation of delivery systems, i.e., Sapien 3, Evolute R and Evolute PRO. Each group was divided into subgroups according to the expandable system (SEV vs. BEV). The RPC was also categorized into two groups, RPC < 30% and RPC ≥ 30%.

### 2.2. Definition Criteria for Events and Thrombocytopenia

Postprocedural events were defined according to the Valve Academic Research Consortium-2 criteria. The lowest recorded platelet count during hospitalization was defined as the nadir platelet count. This formula was used to calculate the reduction in platelet count (RPC): [%RPC = 100 × (baseline platelet count − nadir platelet count)/baseline platelet count].

### 2.3. Statistical Analysis

Mean ± standard deviations are used for continuous variables and for frequencies and percentages for categorical variables. Normality was tested for continues variables between the various study groups via a Shapiro–Wilk test and a Mann–Whitney non-parametric test was performed when abnormal distributions were found. A Pearson’s chi-square test was used for categorical variables when appropriate. The main effect estimates are presented along with their 95% confidence intervals.

The Kaplan–Meier test was used for cumulative survival analysis after six months. For the comparison between patient survival of those with an RPC < 30% and those with an RPC ≥ 30%, a log-rank test was used when appropriate. A *p* value of <0.05 was defined as statistically significant. Analyses were performed with an IBM Statistical Package for the Social Sciences (SPSS) version 23.0 (IBM Corp., Armonk, NY, USA). This study was approved by the Helsinki Committee of Kaplan Medical Center.

## 3. Results

### 3.1. Patients’ Characteristics

Three-hundred and eighty consecutive patients were enrolled during the study period of 10 years. The flowchart for the study can be seen in Figure 1. The final population analyzed included 334 patients. The first-generation valves (Sapien, Sapien XT, CoreValve) were implanted before the year 2016 (133 patients) and the subsequent 201 patients received the newer valves (SAPIEN 3, Evolute-Pro, Evolute-R).

### 3.2. Platelet Count Kinetics

The study population’s baseline and procedural characteristics for the two periods are summarized in Table 1. Patients that received TAVI in the early period had slightly more dyslipidemia and atrial fibrillation. Coronary artery disease prevalence was higher in the contemporary group. The two valve types (SEV vs. BEV) showed no differences between them in the two periods according to baseline and procedural characteristics. A platelet count decrease after TAVI occurred in 95.8% of the patients. In the early period, the mean RPC was 33% ± 15%, while in the contemporary period it was 26% ± 14%.

In the first generation of TAVI, we found that 62% of patients had a platelet count decrease of ≥30% in comparison to 33% in the newer generation (*p* value < 0.01). In both groups, the nadir of RPC was on the third day in comparison with the first day, with a significant *p*-value, <0.01 (Figure 2). However, no differences were found in the time to reach the nadir between the two expandable valve types (*p*-value 0.97 and 0.417, respectively). These patients were followed for six months, and we found a normalization in their platelet count without further sequelae during this period (Table 2).

### 3.3. Variables Analysis

The variables related to elevated RPC after TAVI are presented in Table 3a,b. In the early period, the only factor that related to a ≥30% RPC was the higher left ventricular ejection fraction. In the contemporary period, the procedure time and contrast amount used were related to a ≥30% RPC with a *p*-value of <0.05. Additionally, the residual AV gradient (13.79 ± 6.92) was associated with greater decreases in platelet count (*p*-value 0.057). We found no correlation between the expandable system type and a decrease in platelet count in both periods (*p*-value 0.787 and 0.292, respectively).

### 3.4. Patient Outcomes

During the ten-year study period, there was no significant difference in mortality between the expandable system types (*p*-value = 0.579). In Figure 3, we show the 10-year follow-up Kaplan–Meier survival curve, and in Table 4, we summarize the thirty-day clinical outcomes. Elevated RPC levels at 30 days were associated with higher rates of major bleeding, vascular complications, and mortality.

## 4. Discussion

We studied the actual characterization of TAVI-rTP phenomena across 10 years. This included predictors, incidence and clinical prognostic significance. Our study also differentiated between earlier and newer TAVI generations. Our study’s major findings are: (a) a reduction in the development of severe TAVI-rTP thanks to newer TAVI valves even though there still is a persistent high incidence of TAVI-rTP; (b) no is correlation between TAVI-rTP and the valve type (SEV or BEV TAVI delivery systems); and (c) there is a correlation between MACE at 30 days and severe TAVI-rTP, as is also seen in the literature.

Since first being performed in humans by Alain Cribier in 2002 [18], transcatheter therapy for severe aortic stenosis has undergone rapid growth, with more than 500,000 procedures worldwide [19]. In the last few years, several TAVI-rTP investigations have been conducted, mostly on Corevalves and some on Sapien valves [2,4,11,13], and only a few studies have compared the two types of expandable systems [10,14]. These studies have showed that TAVI-rTP incidence and dynamics have remained unchanged.

In our study, a platelet count decrease after TAVI occurred in more than 90% of our population in both generations. In early generation TAVI, the mean RPC was 33 ± 15%, and in the contemporary generation it was 26 ± 14%. The percentage of patients that developed significant severe thrombocytopenia (i.e., RPC ≥ 30%) in the later period was half of what it was in the earlier period (33% vs. 62%, respectively).

A previous study raised the question about the delivery system factor [14]. They showed that BEV valves were related to higher RPC than SEV valves. This hypothesis was based on differences in the prosthesis design, but the study included only first generation TAVI valves. Our findings include all valve delivery systems, and we found that they have no influence on RPC for both periods.

The newer Medtronic SEV TAVI still preserves several features of the old Corevalve. Newer improvements include: the outflow portion being shortened by 10mm and redesigned to better fit the aortic root, the geometry of the inflow portion’s distal diamond cell being modified with a configuration that is slightly asymmetric and extended in length for better conformability and to ensure consistent radial force across a range of annulus sizes, the nadir of the valve leaflets being sutured from the edge of the inflow portion and the distal skirt being been extended with a scalloped design for better sealing. Instead of an 18 Fr TF delivery sheath, the newer one is delivered via a 14 Fr in-line sheath [20,21].

In 2016, we started using the Edward SAPIEN 3 (S3) (Edwards Lifesciences), which is the fourth generation in the Edwards family of balloon-expandable transcatheter heart valves [22]. The S3 has a modified bovine pericardial tissue leaflet and stent design, further downsizing the overall profile. The Edwards Certitude delivery system has been downsized from the previous 24 Fr Ascendra+ System to a smaller 18–14 Fr. These improvements reduce its profile, improve positioning and deployment and reduce paravalvular regurgitation.

The use of iodinated contrast agents is another possible etiologic factor [2,17]. These contrast agents’ chemical properties, immune-allergic reactions or genetic predispositions, and provide explanations for these relationships. During the early period, no relationship was found between the procedure duration, contrast volume and RPC. However, in the contemporary period, there was a significant correlation. It is possible that the positive correlation found between RPC, the duration of the procedure and contrast agent amount is a surrogate of the procedure’s complexity more than just a mechanistic cause. In more complex cases, tissue damage from the procedure, which is responsible for platelet activation and systemic inflammatory responses, was more pronounced. Team expertise is another important factor. Gallet et al. [2] found a significant relationship between platelet count decrease and MACEs during the learning curve period. In our study, the 380 cases analyzed were done by the same team.

When we look at clinical outcomes, the association between significant post-TAVI RPC and life-threatening or major bleeding, major vascular complications and mortality within 30 days is in agreement with other studies [11,12,13,14]. The TAVI-rTP was more severe during the early period and is associated with worsening clinical outcomes. Since these were more complicated cases, TAVI-rTP can be defined as more of a result than the cause. Our study did not differentiate between worsening pre-existing TAVI-rTP or acquired TAVI-rTP. However, our findings demonstrate that TAVI-rTP can predict and is associated with MACE at 30 days, as in Flaherty et al. [4].

To the best of our knowledge, this is the first report on TAVI-rTP analysis using a complete valve system spectrum over one decade which compares an early and contemporary period. This comparison reflects the evolution of this clinical procedure. Our team’s expertise, population indication extension and improvements in technical procedures can help explain the significant TAVI-rTP reduction, with a null influence of the implantation system on TAVI-rTP.

Our study has certain limitations. One partial limitation is it being a single-center retrospective observational study. However, being that only one team performed the procedure means that treatment uniformity was maintained. An additional limitation is our sample size. Although it is large, it is not robust and therefore did not allow for sufficient capacity to detect overall mortality event rates.

In conclusion, in our cohort, contemporary TAVI-rTP is still a common phenomenon. However, severe thrombocytopenia is significantly less frequent. This can be explained by the team’s expertise, population indication extension and improvements in technical procedures which all contributed to the reduction in significant TAVI-rTP, with no influence of the implantation system on TAVI-rTP. RPC of 30% or more is associated with a poor 30-day outcome.

## Figures and Tables

**Figure 1 jcm-11-00726-f001:**
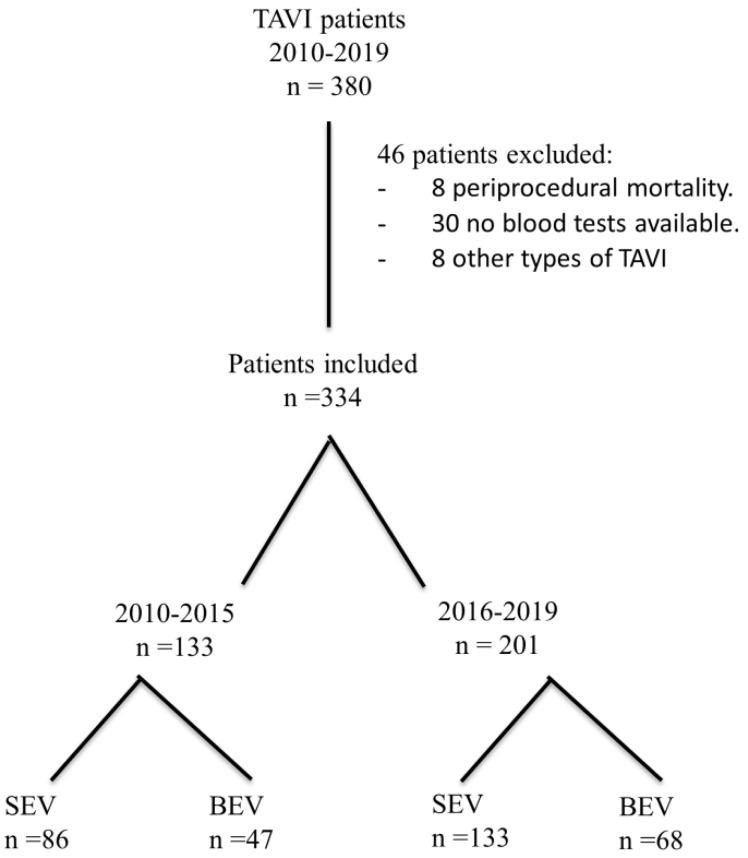
Study flowchart.

**Figure 2 jcm-11-00726-f002:**
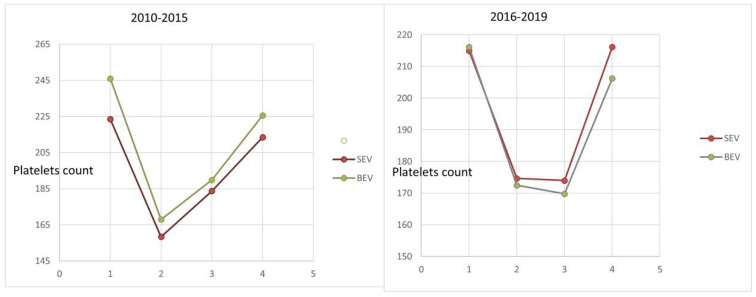
Kinetic behavior of drop platelet count between self-expanding and balloon-expandable valve groups, old versus new generation TAVI.

**Figure 3 jcm-11-00726-f003:**
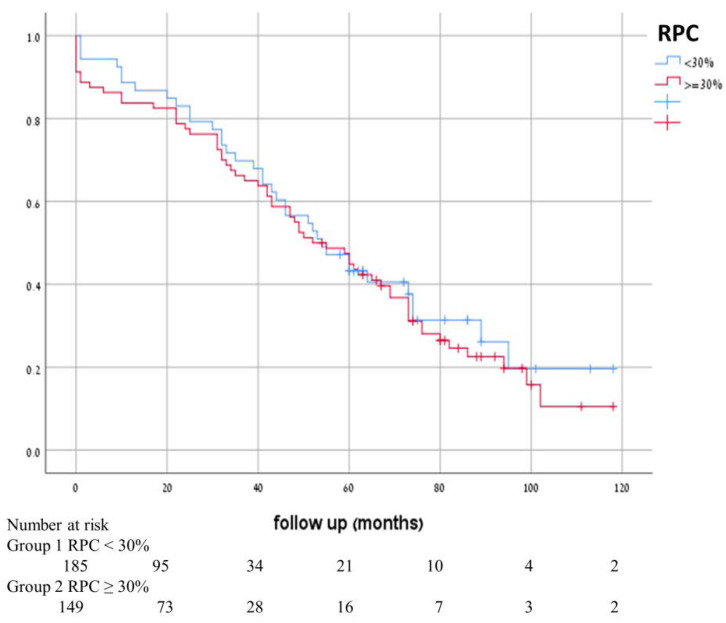
Kaplan–Meier survival curves after transcatheter aortic valve replacement according the percentage of drop platelet count.

**Table 1 jcm-11-00726-t001:** Patient’s baseline characteristics.

	Early (2010–2015)				Contemporaneous (2016–2019)				*p* Value between the Two Groups
	Total	SEV	BEV	*p*-Value	Total	SEV	BEV	*p*-Value	
Number (334)	133	86	47		201	133	68		
Age (years)	82.68 ± 6.33	82.81 ± 6.40	82.45 ± 6.25	0.759	81.14 ± 7.12	81.05 ± 7.52	81.33 ± 6.29	0.789	0.045
BMI (kg/m²)	27.84 ± 5.44	27.59 ± 5.12	28.28 ± 5.99	0.492	28.42 ± 4.95	28.25 ± 5.16	28.77 ± 4.52	0.479	0.314
BSA (m²)	1.76 ± 0.20	1.76 ± 0.20	1.78 ± 0.20	0.535	1.80 ± 0.18	1.80 ± 0.18	1.81 ± 0.17	0.752	0.06
Gender (male)	36.8%	36%	38.3%	0.794	47.8%	47.4%	48.5%	0.876	0.049
Hypertension	93.2%	94.2%	91.5%	0.554	91%	88.7%	95.6%	0.107	0.473
Diabetes	39.8%	34.9%	48.9%	0.114	42.5%	44.7%	38.2%	0.381	0.631
Dyslipidemia	55.6%	53.5%	59.6%	0.449	88.6%	88%	89.7%	0.715	<0.0001
Smoker	9.8%	7%	14.9%	0.142	10%	12.1%	5.9%	0.164	0.964
Atrial fibrillation	38%	35.4%	42.6%	0.418	23.4%	24.8%	20.6%	0.503	0.004
CAD	27.5%	32.9%	17.4%	0.057	46.8%	48.1%	44.1%	0.591	<0.0001
PAD	11.5%	10.6%	13%	0.674	14.4%	15%	13.2%	0.731	0.434
Previous myocardial infarction	9.8%	14.1%	2.1%	0.027	9.5%	11.3%	5.9%	0.216	0.905
Previous CVA	4.5%	7%	1%	0.064	10%	11.3%	7.4%	0.379	0.069
Pacemaker	9%	9.3%	8.5%	0.879	14.9%	15%	14.7%	0.95	0.111
CABG	20%	20%	17%	0.677	7%	5.3%	10.3%	0.185	0.268
STS	9.28 ± 2.34	9.47 ± 2.37	9.2 ± 2.62	0.394	8.01 ± 1.5	8.04 ± 1.45	7.95 ± 1.62	0.617	0.136
LVEF (%)	53.72 ± 8.31	53.47 ± 8.21	54.19 ± 8.57	0.637	52.86 ± 9.82	52.52 ± 10.10	53.52 ± 9.30	0.501	0.405
AVA (cm^2^)	0.66 ± 0.14	0.64 ± 0.13	0.68 ± 0.16	0.213	0.70 ± 0.16	0.68 ± 0.16	0.74 ± 0.15	0.023	0.028
AV gradient	76.24 ± 22.29	76.17 ± 21.42	76.38 ± 24.03	0.595	74.97 ± 22.62	75.25 ± 23.43	74.42 ± 21.09	0.809	0.615
Time (min)	107.87 ± 36.58	107.02 ± 36.31	109.4 ± 37.44	0.717	84.24 ± 26.79	85.28 ± 29.03	82.16 ± 21.69	0.398	<0.0001
Contrast volume (mL)	169.26 ± 61.71	168.69 ± 65.55	170.2 ± 54.66	0.887	113.6 ± 48.58	118.4 ± 50.68	104.1 ±42.95	0.05	<0.0001

Abbreviations: AVA, aortic valve area; AV, aortic valve; BMI, body mass index; BSA, body surface area; CABG, coronary artery bypass; CAD, coronary artery disease; CVA, cerebrovascular accident; LVEF, left ventricle ejection fraction; PAD, peripheral arterial disease.

**Table 2 jcm-11-00726-t002:** Platelet count (10^9^/L) from baseline to six-month follow up.

Time	2010–2015	SEV	BEV	*p* Value	2016–2019	SEV	BEV	*p* Value
Baseline	227.4 ± 97.2	220.1 ± 71.5	241.0 ± 131.9	0.237	212.5 ± 72.7	214.9 ± 72.0	216.1 ±78.4	0.258
Day 1	158.4 ± 78.4	156.2 ± 57.2	162.5 ± 107.5	0.663	172.1 ± 62.6	174.6 ± 63.6	172.5 ± 65.5	0.732
Day 3	182.0 ± 75.3	179.3 ± 60.1	187.0 ± 98.1	0.579	169.7 ± 62.5	174.0 ± 65.4	169.8 ± 59.1	0.488
6 months	217.5 ± 73.4	213.5 ± 76.0	224.4 ± 81.3	0.458	213.0 ± 76.7	216.2 ± 76.0	206.8 ± 77.0	0.332

Abbreviations: BEV, balloon-expandable valves; SEV, self-expandable valves.

**Table 3 jcm-11-00726-t003:** Variable analysis of factors related to high RPC for early group (2010–2015) and contemporaneous group (2016–2019). (**a**) Variable analysis of factors related to high RPC for early group (2010–2015). (**b**) Variable analysis of factors related to high RPC for contemporaneous group (2016–2019).

**(a)**				
**Variable**	**Total** **(*n* = 133)**	**RPC < 30%** **(*n* = 51)**	**RPC ≥ 30%** **(*n* = 82)**	***p* Value**
Age (years)	82.7 + 6.3	82.9 ± 6.2	82.6 ± 6.4	0.948
BMI (kg/m^2^)	27.8 ± 5.4	26.45 ± 4.39	28.73 ± 5.87	0.038
Gender (male)	36.8%	45.3%	31.3%	0.100
Hypertension	93.2%	90.6%	95.0%	0.319
Diabetes mellitus	39.8%	35.8%	42.5%	0.443
Dyslipidemia	55.6%	60.4%	52.5%	0.371
Smoker	9.8%	7.5%	11.3%	0.481
Atrial fibrillation	38.0%	32.0%	41.8%	0.265
CAD	27.5%	31.4%	25.0%	0.426
PAD	11.5%	11.3%	11.5%	0.969
Previous MI	9.8%	7.7%	11.3%	0.503
Previous CVA-TIA	4.5%	3.8%	5.0%	0.739
Pacemaker	9.0%	7.5%	10.0%	0.629
Previous CABG	20.0%	0.0%	25.0%	0.576
STS Score	9.28 ± 2.34	9.31 ± 2.23	9.25 ± 2.61	0.462
TAVI types (SEV)	64.7%	66.0%	63.8%	0.787
LVEF (%)–pre TAVI	53.7 ± 8.3	52.06 ± 8.53	54.84 ± 8.04	0.040
AVA (cm^2^)–pre TAVI	0.66 ± 0.15	0.67 ± 0.15	0.66 ± 0.14	0.744
AV GRADIENT (mm Hg)–post TAVI	76.25 ± 22.3	76.00 ± 24.65	76.41 ± 20.74	0.636
Contrast volume (mL)	169.3 ± 61.72	173.85 ± 66.71	166.23 ± 58.41	0.730
Time (min)	107.9 ±36.6	103.96 ± 32.55	110.49 ± 39.05	0.546
**(b)**				
**Variable**	**Total** **(*n* = 201)**	**RPC < 30%** **(*n* = 134)**	**RPC ≥ 30%** **(*n* = 67)**	***p* Value**
Age (years)	81.1 ± 7.1	80.57 ± 7.56	82.30 ± 6.03	0.159
BMI (kg/m^2^)	28.1 ± 5.4	28.18 ± 4.73	28.96 ± 5.44	0.339
Gender (male)	47.8%	47.1%	48.9%	0.643
Hypertension	91.0%	86.7%	92.6%	0.152
Diabetes mellitus	42.5%	42.7%	40.2%	0.281
Dyslipidemia	88.6%	87.0%	90.7%	0.614
Smoker	10.0%	10.1%	7.9%	0.164
Atrial fibrillation	23.4%	24.2%	20.9%	0.403
CAD	46.8%	47.1%	45.1%	0.641
PAD	14.4%	12.0%	16.2%	0.362
Previous MI	9.5%	10.3%	6.9%	0.216
Previous CVA-TIA	10.0%	9.3%	8.4%	0.379
Pacemaker	14.9%	13.3%	15.7%	0.550
Previous CABG	7.0%	6.3%	9.6%	0.285
STS Score	8.01 ± 1.5	8.14 ± 1.25	7.80 ± 1.81	0.438
TAVI types (SEV)	68.7%	66.2%	61.2%	0.292
LVEF (%)–pre TAVI	52.8 ± 9.8	51.37 ± 9.94	53.43 ± 7.83	0.248
AVA (cm^2^)–pre TAVI	0.70 ± 0.16	0.71 ± 0.16	0.70 ± 0.16	0.513
AV GRADIENT (mm Hg)–post TAVI	74.97 ± 22.62	74.14 ± 22.83	76.62 ± 22.28	0.057
Contrast volume (mL)	114.0 ± 48.7	108.40 ± 44.17	125.42 ± 55.46	0.028
Time (min)	84.2 ±26.8	79.36 ± 21.94	94.47 ± 32.75	0.002

Abbreviations: AVA, aortic valve area; AV, aortic valve; BMI, body mass index; CAD, coronary artery disease; CVA, cerebrovascular accident; CABG, coronary artery bypass; RPC, reduced platelet count; MI, myocardial infarction; LVEF, left ventricle ejection fraction; PAD, peripheral arterial disease; TAVI, transcatheter aortic valve implantation.

**Table 4 jcm-11-00726-t004:** Thirty-day outcomes of patients with high RPC after TAVI For two consecutive periods.

Variable	Total 2010–2015 *n*= 133	RPC < 30% *n* = 53	RPC ≥ 30% *n* = 80	*p* Value	Total 2016–2019 (*n* = 201)	RPC < 30% (*n*= 134)	RPC ≥ 30% (*n* = 67)	*p* Value	*p* Value between the Two Generations
Life-threatening/major bleeding	18 (13.5%)	4 (7.5%)	14 (17.5%)	0.100	10 (5.0%)	2 (1.5%)	8 (11.9%)	0.001	0.010
Major vascular complication	13 (9.8%)	7 (13.2%)	6 (7.5%)	0.278	18 (9.0%)	7 (5.2%)	11 (16.4%)	0.009	0.952
Acute kidney injury	13 (9.8%)	7 (13.2%)	6 (7.5%)	0.278	7 (3.5%)	4 (3.0%)	3(4.5%)	0.586	0.033
Stroke	6 (4.5%)	2 (3.8%)	4 (5.0%)	0.756	9 (4.5%)	5 (3.7%)	4 (6.0%)	0.469	0.8096
Myocardial infarction	3 (2.3%)	2 (3.8%)	1 (1.3%)	0.337	2 (1.0%)	2 (1.5%)	0 (0.0%)	0.315	0.639
Mortality	7 (5.3%)	0 (0.0%)	7 (8.8%)	0.027	2 (1.0%)	0 (0.0%)	2 (3.0%)	0.044	0.044

Abbreviations: RPC, reduced platelet count; TAVI, transcatheter aortic valve replacement.

## Data Availability

All relevant data are included within the manuscript. All other data presented in this study are available upon request from the corresponding author.

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
