# Peer review of "Trends in Transfemoral Aortic Valve Implantation Related Thrombocytopenia"

_jcm, 2022, doi:10.3390/jcm11030726_

Round 1

Reviewer 1 Report

  • I agree with authors with the use of early vs. Contemporeanous  interventions instead of 1st vs last generation valve comparison. Authors adequately underlined the differences between the proceudures between the 2 groups.
  • Numbers should be added in the KM curve
  • Could authors add the duration of hospitalization? It could be an an important point to emphasize as thrombocytopenia may delay hospital discharge.

Author Response

Reviewer 1.

1) Numbers should be added in the KM curve The figure was upgraded, the number at risk to the KM curve was added (figure 3)

2) Could authors add the duration of hospitalization? The median hospitalization stay was 3. days. All patients admitted evening before the procedure and discharge at POD2 in the majority (91.3%) of the cases . In our sub-analysis, we don’t find a correlation between the RPC and the hospitalization period.

Reviewer 2 Report

 Trends in Transfemoral Aortic Valve Implantation Related 2 Thrombocytopenia

In this study, a total of 334 patients that underwent TAVI between 2010 and 2019 in a single center were analyzed regarding the development of thrombocytopenia following TAVI. Patients with periprocedural death were excluded from analysis. The main finding is that in the early period of TAVI, RPC was more pronounced than in the contemporary period. An RPC of 30% or higher was associated with bleeding, vascular complications and death.

In the earlier period, the mean RPC was 33 ±15 %, and in the 14 contemporary period (2016-2019) it was 26 ± 14%. In the early group, we found that 62% of the 15 patients had decreased platelet counts of more or equal to 30% in comparison to 33% in the current 16 days. The time of the procedure and the amount of the contrast that has been used in the late period 17 were associated with significant RPC (p value 0.002 and 0.028 respectively).

Even though the topic is of interest, there are many issues that need to be addressed by the authors. Here are my thoughts:

  • Please elaborate on the treatment and proportion of patients who underwent PCI prior to TAVI. Was dual or triple therapy employed?
  • What was the proportion of NOACs? In particular, the strategy of bridging for patients on OAC or NOAC would be interesting.
  • Figure 1, Table 1, Figure 2: it is not helpful and distracting to compare SEV vs. BEV, which creates unnecessary subgroups (multiple testing) as a comparison of SEV vs. BEV was not in the scope of this study, especially as the THV type had no impact on RPC.
  • Table 3a and 3b: I would replace these two tables by one multivariable analysis to identify independent predictors of relevant thrombocytopenia. Instead of separating the early and contemporary group, I would analyze the entire time spectrum and use early vs contemporary experience as co-variate.
  • Figure 3: The Kaplan-Meier curve has multiple issues: what was the median follow up time? If the authors included patients between 2010 and 2019, I assume a large proportion not to be followed up long term. Furthermore, the logrank test should be specified as well as the number at risk.
  • “The newer Medtronic SEV TAVI still preserves several features of the old Corevalve, including trileaflet porcine pericardial tissue, the supra-annular position of the valve leaf-169 lets, and the self-expanding nitinol stent frame. Its improvements include having the outflow portion shortened by 10mm and redesigned to better fit the aortic root, the geometry of the inflow portion’s distal diamond cell is modified with a configuration that is slightly asymmetric and extended length for better conformability and consistent radial force across a range of annulus sizes, the nadir of the valve leaflets is sutured from the edge of the inflow portion and the distal skirt has been extended with a scalloped design for better sealing. Instead of an 18 Fr TF delivery sheath, the newer one is delivered via a 14 Fr in line sheath. The prosthesis is fully repositionable and retrievable before a final detachment of the hooks, achieving more controlled and accurate positioning even in difficult anatomies which reduce the risk of paravalvular regurgitation and other unfavorable complications. In 2016, we started using the Edward SAPIEN 3 (S3) (Edwards Lifesciences) which is the fourth generation in the Edwards family of balloon-expandable transcatheter heart valves. The S3 has a modified bovine pericardial tissue leaflet and stent design, further 183 downsizing the overall profile. The Edwards Certitude delivery system has been down-184 sized from the previous 24 Fr Ascendra+ System to a smaller 18 - 14 Fr. It also has a central radiopaque marker and an additional fine alignment wheel, providing accurate and more reproducible S3 positioning. These improvements reduce its profile, improve positioning and deployment, and reduce paravalvular regurgitation.”

This paragraph is superfluous and should be either shortened or deleted. Currently, it looks like an advertisement of the manufacturers. If shortened please put the content in relation to the findings of the present study.

  • The discussion on the role of contrast agent is appropriate and comprehensive, even though it is highly theoretical. I agree that probably the relation between contrast agent amount and thrombopenia is rather a surrogate of the procedure´s complexity, including more manipulation and coagulation activation.
  • Please check the references, some of them are incomplete.

Author Response

1) Please elaborate on the treatment and proportion of patients who underwent PCI prior to TAVI. Was dual or triple therapy employed?

The patients were treated according to ESC guidelines recommendation in each period. Patients with atrial fibrillation (nearly 30% of our population) or not have. The guidelines changed across the time, at the beginning they suggest treatment with DAPT after TAVI and then after they suggest SAPT.

2) What was the proportion of NOACs? In particular, the strategy of bridging for patients on OAC or NOAC would be interesting.

According to ESC guidelines, we don’t practice the bridging strategy before percutaneous procedures.

3) Figure 1, Table 1, Figure 2: it is not helpful and distracting to compare SEV vs. BEV, which creates unnecessary subgroups (multiple testing) as a comparison of SEV vs. BEV was not in the scope of this study, especially as the THV type had no impact on RPC

In our study the different deployment systems types does not have impact. We decide to include this information due to in other studies they found an impact of the delivery system on RPC.

4) Table 3a and 3b: I would replace these two tables by one multivariable analysis to identify independent predictors of relevant thrombocytopenia.

Instead of separating the early and contemporary group, I would analyze the entire time spectrum and use early vs contemporary experience as co-variate. The essence in our study is the comparison between the “old” TAVI generation and the “ newer” delivery system one.

5) Figure 3: The Kaplan-Meier curve has multiple issues: what was the median follow up time? If the authors included patients between 2010 and 2019, I assume a large proportion not to be followed up long term. Furthermore, the long rank test should be specified as well as the number at risk.

We added the number at risk to KM curve, the median follow up is nearly 5 years but this is not the issues , the RPC occur in the acute phase and the composite outcome ( MACE ) is generally at short term.

6) “The newer Medtronic SEV TAVI still preserves several features of the old Corevalve, including trileaflet porcine pericardial tissue, the supra-annular position of the valve leaflets, and the self-expanding nitinol stent frame. Its improvements include having the outflow portion shortened by 10mm and redesigned to better fit the aortic root, the geometry of the inflow portion’s distal diamond cell is modified with a configuration that is slightly asymmetric and extended length for better conformability and consistent radial force across a range of annulus sizes, the nadir of the valve leaflets is sutured from the edge of the inflow portion and the distal skirt has been extended with a scalloped design for better sealing. Instead of an 18 Fr TF delivery sheath, the newer one is delivered via a 14 Fr in line sheath. The prosthesis is fully repositionable and retrievable before a final detachment of the hooks, achieving more controlled and accurate positioning even in difficult anatomies which reduce the risk of paravalvular regurgitation and other unfavorable complications. In 2016, we started using the Edward SAPIEN 3 (S3) (Edwards Lifesciences) which is the fourth generation in the Edwards family of balloon-expandable transcatheter heart valves. The S3 has a modified bovine pericardial tissue leaflet and stent design, further downsizing the overall profile. The Edwards Certitude delivery system has been down sized from the previous 24 Fr Ascendra+ System to a smaller 18 - 14 Fr. It also has a central radiopaque marker and an additional fine alignment wheel, providing accurate and more reproducible S3 positioning. These improvements reduce its profile, improve positioning and deployment, and reduce paravalvular regurgitation.” This paragraph is superfluous and should be either shortened or deleted. Currently, it looks like an advertisement of the manufacturers. If shortened please put the content in relation to the findings of the present study.

This paragraph was shortened, we preserved the description of the newer relevant developments can explain the decrease in severe thrombocytopenia post TAVI

7) Please check the references, some of them are incomplete.

We recheck the references and we reenter them by the “endnotes” program.

This manuscript is a resubmission of an earlier submission. The following is a list of the peer review reports and author responses from that submission.